# Auto DragGAN: Editing the Generative Image Manifold in an Autoregressive Manner

Submission Id: 471

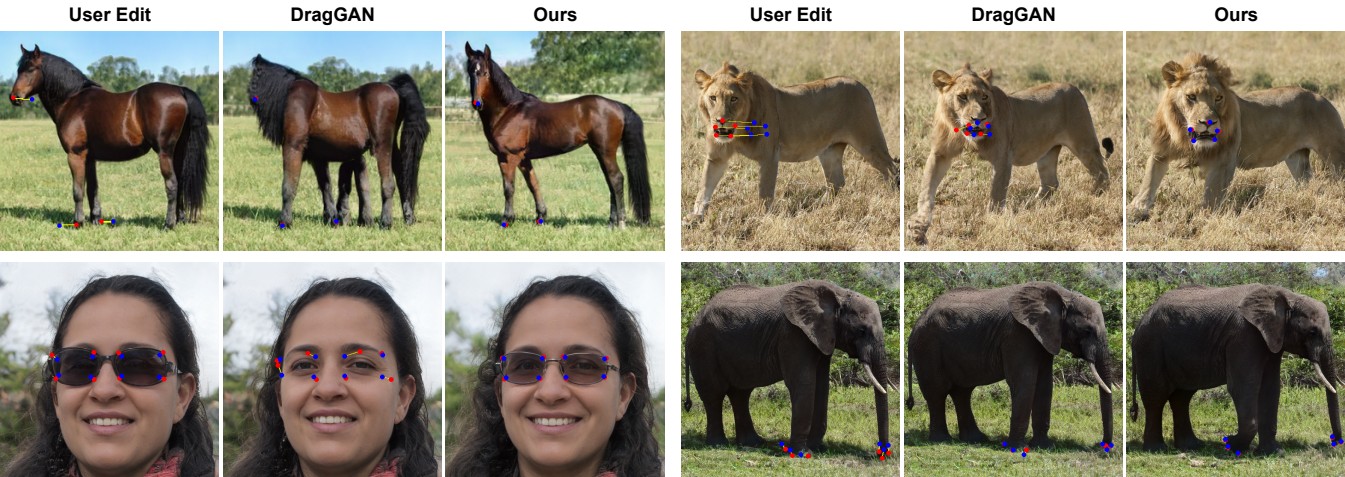

**Figure 1: Users are able to specify handle points (marked as red) and target points (marked as blue) on any GAN-generated images, and our method will precisely move the handle points to reach their corresponding target points, thereby achieving the desired drag effect on the image. We compare DragGAN [29] with our proposed Auto DragGAN, where our method demonstrates superior drag performance.**

## ABSTRACT

Pixel-level fine-grained image editing remains an open challenge. Previous works fail to achieve an ideal trade-off between control granularity and inference speed. They either fail to achieve pixel-level fine-grained control, or their inference speed requires optimization. To address this, this paper for the first time employs a regression-based network to learn the variation patterns of Style-GAN latent codes during the image dragging process. This method enables pixel-level precision in dragging editing with little time cost. Users can specify handle points and their corresponding target points on any GAN-generated images, and our method will move each handle point to its corresponding target point. Through experimental analysis, we discover that a short movement distance from handle points to target points yields a high-fidelity edited image, as the model only needs to predict the movement of a small portion of pixels. To achieve this, we decompose the entire movement process into multiple sub-processes. Specifically, we develop

a transformer encoder-decoder based network named 'Latent Predictor' to predict the latent code motion trajectories from handle points to target points in an autoregressive manner. Moreover, to enhance the prediction stability, we introduce a component named 'Latent Regularizer', aimed at constraining the latent code motion within the distribution of natural images. Extensive experiments demonstrate that our method achieves state-of-the-art (SOTA) inference speed and image editing performance at the pixel-level granularity.

## CCS CONCEPTS

• **Computing methodologies → Computer vision**; **Image manipulation**.

## KEYWORDS

GANs, Image Editing, Autoregressive Model

## 1 INTRODUCTION

Significant advances [12, 15, 23, 45] in the field of image generation have also fostered research in image editing. Images obtained by generative models [18–20, 35] can now satisfy the needs of most users, yet they lack flexible and free control. Editing [21, 29] images generated by these models can provide users with the flexible and free control they desire, thereby enabling them to obtain images that meet their specific requirements. Image editing methods based on generative models have attracted widespread attention

among researchers. However, fine-grained control in image editing remains an open challenge, especially at the pixel level. Previous research [3, 14, 21, 27, 29, 31, 34] has failed to achieve an ideal trade-off between control granularity and inference time. They either failed to achieve pixel-level fine-grained control, or their inference speed required optimization. Numerous methods [14, 21, 27, 31] enable image editing based on text prompts, with editing operations including replacing image subjects, modifying subject poses, and altering image styles. Additionally, PTI [34] leverages attribute labels to guide StyleGAN [17–20] in modifying specific attributes of images, such as facial expressions, face orientations, and the age of persons. Due to the limitations of text prompts and attribute labels in delivering fine-grained information, these methods are restricted to coarse-grained control. Currently, most research focuses on coarse-grained control, thus fine-grained image editing still has many problems to be investigated and solved.

The user can annotate a StyleGAN [17–20] image with locations they want to move and specifies a movement direction by mouse dragging. From these user inputs and initial latent codes, UserControllableLT [8] estimates the output latent codes, which are fed to the StyleGAN [17–20] generator to obtain a result image. While UserControllableLT [8] provides users with significantly greater editing flexibility compared to previous methods, it still fails to achieve pixel-level control.

Recently, DragGAN [29] has achieved an interactive image editing method based on pixel manipulation for the first time, which allows users to drag the image subject. This method has resulted in astonishing drag editing effects with pixel-level precision. However, the principal idea of DragGAN [29] is the iterative reverse optimization of latent codes in the StyleGAN [17–20] space, which requires enhancement in computational efficiency.

Overall, previous research has failed to achieve an ideal trade-off between control granularity and inference speed. They either fail to achieve pixel-level fine-grained control, or their inference speed requires optimization. To do this, this paper introduces a regression based network for learning the variation patterns of latent codes within the StyleGAN [17–20] space during the image dragging process, thereby achieving pixel-level precision in drag editing with little time cost. Compared to UserControllableLT [8], our method achieves pixel-level fine-grained control. In comparison with DragGAN [29], our approach achieves a comparable level of fine-grained control while significantly reducing the required inference time. As illustrated in Figure 2, our method achieves an ideal trade-off between control granularity and inference speed. Extensive experiments demonstrate that our method achieves state-of-the-art (SOTA) inference speed and image editing performance at the pixel-level granularity.

We develop a transformer encoder-decoder based network named 'Latent Predictor' to predict the latent code motion trajectories from handle points to target points in an autoregressive manner. Moreover, to enhance the prediction stability, we introduce a component named 'Latent Regularizer', aimed at constraining the latent code motion within the distribution of natural images.

Specifically, we propose a two-stage training strategy. In the first stage, we introduce the Latent Regularizer to constrain the latent code motion, ensuring that the latent code remains within the reasonable distribution of the StyleGAN [17–20] latent space to

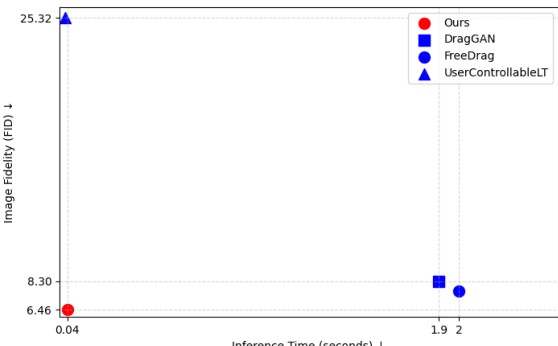

Figure 2: The comparison between UserControllableLT [8], DragGAN [29], FreeDrag [26] and our proposed Stable Drag-GAN in terms of key performance indicators. Inference time (seconds) ↓ and image fidelity (FID) ↓ were both tested in the face landmark manipulation experiment under the settings described in Section 4.3.1, based on the 'one point' setting.

enhance the stability of the Latent Predictor. By introducing random noise to the latent codes, we generate outlier latent codes that fall outside the reasonable distribution of the StyleGAN [17–20] latent space. Subsequently, we train the Latent Regularizer utilizing an attention mechanism to learn the internal structural information within the latent code, thereby correcting outlier latent codes back within the reasonable distribution of the StyleGAN [17–20] latent space.

In the second stage of training, we have developed a network based on the transformer encoder-decoder architecture, which we refer to as the 'Latent Predictor'. This network effectively converts the image drag problem [29] into a latent code motion sequences regression task. It is jointly trained with the Latent Regularizer to regularize the prediction results. Initially, to obtain pseudo-labels for training, we introduce continuous and slight random noise into the randomly sampled latent codes to generate latent code motion sequences. These motion sequences simulate a 'pseudo-process' to approximate the actual dragging process. The Latent Predictor autoregressively predicts this 'pseudo-process', employing a cross-attention mechanism to learn the motion trajectories from handle points to target points, thereby precisely moving the handle points to their corresponding target points.

In summary, the three principal contributions of this paper are as follows: (1) For the first time, we present a regression-based network that achieves pixel-level fine-grained image editing; (2) We convert the image dragging problem into a regression problem of latent code motion sequences for the first time and propose a Latent Regularizer as well as a Latent Predictor based on a transformer encoder-decoder architecture; (3) Extensive experiments demonstrate the effectiveness and efficiency of our method, which achieves an ideal trade-off between control granularity and inference speed. Extensive experiments demonstrate that our method achieves state-of-the-art (SOTA) inference speed and image editing performance at the pixel-level granularity.

## 2 RELATED WORK

### 2.1 Generative Models

**GANs**. Generative Adversarial Networks (GANs) are a class of generative models that function by transforming low-dimensional, randomly sampled latent vectors into realistic images. These models employ adversarial learning for training and have been demonstrated to be capable of generating high-resolution, lifelike images [6, 12, 17–20]. However, most GAN models, such as StyleGAN [17–20], do not support direct and controllable editing of the generated images in their original design. To overcome this limitation, several methods have been proposed to condition Generative Adversarial Networks (GANs). In these approaches, the network receives not only randomly sampled latent vectors but also conditional inputs, such as segmentation maps [16, 30] or 3D variables [7, 11], to generate realistic images. EditGAN [25] achieves image editing by first modeling the joint distribution of images and segmentation maps, followed by computing a new image corresponding to the edited segmentation map.

**Diffusion Models**. Recently, diffusion models [41] have been demonstrated to be capable of high-quality image synthesis [15, 42, 43]. These models iteratively denoise randomly sampled noise to create realistic images. The latest models have shown the potential for expressive image synthesis conditioned on text inputs [33, 35, 36]. However, natural language inputs lack the ability to finely control the spatial attributes of images, thus limiting all text-conditioned methods to high-level semantic editing. Additionally, current diffusion models are slower in synthesizing images due to their requirement for multiple denoising steps. Despite progress in efficient sampling, Generative Adversarial Networks (GANs) still hold an advantage in terms of efficiency.

### 2.2 Generative Models for Interactive Content Creation

Several methods have been proposed for editing unconditional Generative Adversarial Networks (GANs) by manipulating the input latent vectors. Some approaches rely on supervised learning from manually annotated or existing 3D models to discover meaningful latent directions [1, 24, 32, 37, 44]. Others identify significant semantic directions in the latent space in an unsupervised manner [13, 37, 38, 49]. Recently, control over the coarse positioning of objects has been achieved by introducing intermediate "blobs" [9] or heatmaps [47]. All these methods allow for the editing of semantic attributes in images, such as appearance, or coarse geometric properties, like object positioning and pose. Although Editing-in-Style [5] has demonstrated some capability in spatial attribute editing, it achieves this solely through the transfer of local semantics between different samples.

### 2.3 Points-based for Interactive Content Creation

UserControllableLT [8] and DragGAN [29] are point-based editing methods that have been previously proposed. Particularly, Drag-GAN [29] allows users to input handle points and target points, enabling the dragging manipulation of images. Concurrent to our

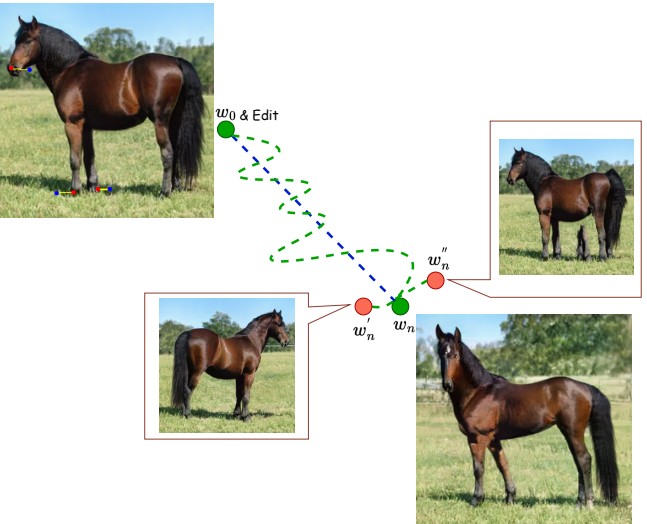

**Figure 3: The outlier latent codes. The shortest motion path in the $\mathcal{W}^+$ space between the latent code $w_0$ and its edited result $w_n$ is depicted as the blue dashed line in the figure, while the green dashed line represents the motion trajectory learned by our model. $w_n^{'}$ and $w_n^{''}$ are the outlier latent codes, predicted by the model without the use of the Latent Regularizer.**

work are FreeDrag [26] and DragDiffusion [39]. FreeDrag [26] proposes a novel point-tracking-free paradigm to enhance DragGAN [29]. DragDiffusion [39] extends the editing framework of Drag-GAN [29] to diffusion models. DragGAN [29], FreeDrag [26], and DragDiffusion [39] are all methods based on the optimization of latent codes. Our proposed method differs significantly from all of these approaches.

## 3 METHOD

In this paper, we propose a novel regression-based network architecture that achieves fine-grained image editing at the pixel level. Given a source image and its handle points and target points, the network predicts the motion trajectories in the StyleGAN latent space to make the handle points reach their corresponding target point positions in image space. Initially, in Section 3.1, we briefly introduce the preliminaries of StyleGAN. Subsequently, in Section 3.2, we introduce the Latent Regularizer, aimed at constraining the latent code motion within a reasonable range. In Section 3.3, the Latent Predictor, which is employed to predict the latent code motion sequences, is discussed.

### 3.1 Preliminaries of StyleGAN

In StyleGAN2 [20], the mapping network takes a 512-dimensional latent code $z$ from a normal distribution and maps it to an intermediate latent code $w$ in a 512-dimensional space. This space is referred to as the $\mathcal{W}$ space. The generator network then uses $w$, either a single value or multiple distinct values for different layers, to produce the output image. The process involves copying $w$ several times, sending it to various generator layers, thereby controlling

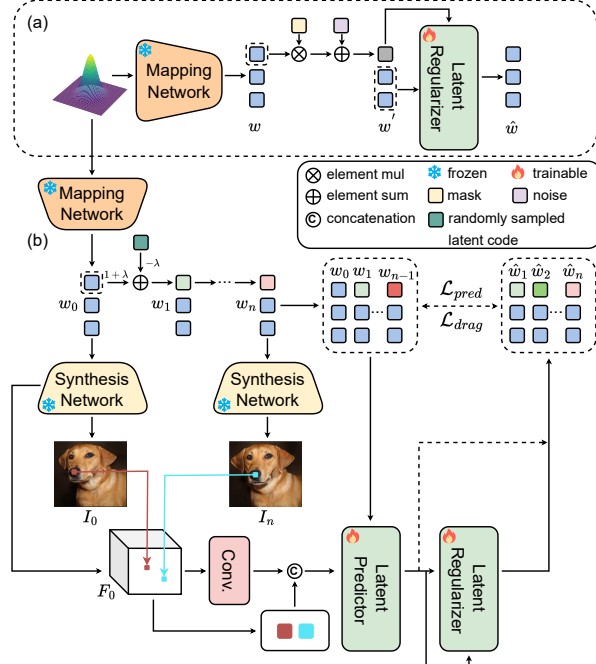

**Figure 4: The overview of our proposed Auto DragGAN. (a) corresponds to the first stage of training, namely the pre-training of the Latent Regularizer. (b) represents the second stage of training, which is the joint training of the Latent Predictor and the Latent Regularizer.**

different image attributes. The dimension of $w$ can be extended to $l \times 512$ in the $\mathcal{W}^+$ space, where $l$ is the number of layers, offering more expressiveness. This advanced architecture allows for more precise control over the generated images, enhancing the quality and reducing artifacts. For a detailed technical explanation, please refer to the original paper on StyleGAN2 [20]. Our work is based on the $\mathcal{W}^+$ space.

## 3.2 Latent Regularizer

Our training process is divided into two distinct stages. The first stage is dedicated to the pre-training of the Latent Regularizer, followed by the second stage which focuses on the joint training of both the Latent Predictor and the Latent Regularizer. This section will detail the training conducted during the first stage as well as the proposed Latent Regularizer.

As illustrated in Figure 3, due to the complex distribution of the $\mathcal{W}^+$ space, even minor inference errors can generate outlier latent codes that fall outside the reasonable distribution of the $\mathcal{W}^+$ space. This can lead to a significant degradation in the fidelity of the generated images, which is manifested as artifacts or incorrect dragging in the pixel space. Therefore, we need to train an additional Latent Regularizer to ensure that the latent code motion remains within the reasonable distribution of the $\mathcal{W}^+$ space. This assists the Latent Predictor in more stably forecasting the latent code motion sequences.

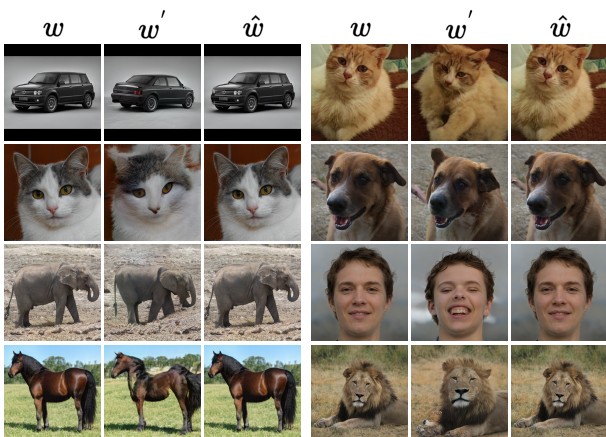

**Figure 5: Reconstruction of the outlier latent codes. For each set of images, the first, second, and third columns correspond to the initial random sampled latent code $w$, the outlier latent code $w'$, and the reconstructed $\hat{w}$, respectively.**

As illustrated in Figure 4 (a), the mapping network of StyleGAN2 [20] randomly samples a 512-dimensional latent code $z$ from a normal distribution and maps it to a latent code $w$ of dimension $l \times 512$, where $l$ represents the number of layers in the generator network. These mapped latent codes serve as training samples for the Latent Regularizer. UserControllableLT [8] finds that manipulating latent codes on deep layers enables spatial control, such as pose and orientation. DragGAN [29] considers the feature maps after the 6th block of StyleGAN2 [20], which performs the best among all features due to a good trade-off between resolution and discriminativeness. Inspired by UserControllableLT [8] and DragGAN [29], our work is based on the editing of the first six layers of the latent code $w$.

To obtain the outlier latent code, we introduce random noise to the randomly sampled latent code $w$. In the first stage of training, we initially add noise to the first six layers of $w$. Specifically, we perform a masking operation on the first six layers of $w$, randomly setting the vector values of these layers to zero with a 25% probability, followed by the addition of Gaussian noise.

$$w' = (w \odot M) + N \tag{1}$$

where $w'$ represents the outlier latent code, $\odot$ denotes the Hadamard product, $M$ is the masking vector with elements being 0 or 1, and $N$ is the noise sampled from a Gaussian distribution.

Prior work [8, 17–20, 29] finds that manipulating the first six layers of the latent codes enables spatial control, such as pose and orientation. Thus, $w'$ is divided into two sets of vectors: the noisy vectors $w'_1$ from the first six layers and the remaining clean vectors $w'_2$.

Given that $w'_1$ is more closely associated with local features, and $w'_2$ predominantly relates to global features [8, 17–20, 29], we aim to restore the noise-added local features $w'_1$ by leveraging the clean global features $w'_2$.

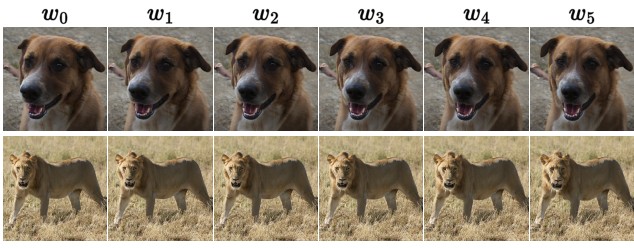

$$w_0 \qquad w_1 \qquad w_2 \qquad w_3 \qquad w_4 \qquad w_5$$

**Figure 6: Visualization of the latent code motion sequence. Given an initial latent code $w_0$, a sequence $w_0, w_1, ..., w_5$ can be generated through the perturbation process described by Equation (7), where $i = 1, 2, 3, 4, 5$.**

The Latent Regularizer structure adopts a standard transformer architecture, with $w_1'$ serving as the key and value for the cross-attention mechanism, while $w_2'$, after being mapped through an MLP to reduce token length, acts as the query for the cross-attention mechanism. The output of the cross-attention mechanism, serving as the restored local features, is concatenated with the clean global features $w_2'$ to form the reconstructed latent code $\hat{w}$.

$$q = Q \cdot (MLP(w_2') \oplus PE) \tag{2}$$

$$k = K \cdot MLP(w_1') \tag{3}$$

$$v = V \cdot MLP(w_1') \tag{4}$$

$$\hat{w} = [(softmax(q \cdot k^T) \cdot v), w_2'] \tag{5}$$

where $PE$ denotes the position embedding, $\oplus$ denotes element-wise sum, and $[,]$ indicates the concatenation operation.

The Latent Regularizer learns to recover clean latent codes from noisy latent codes through a reconstruction task. The reconstruction loss is chosen to be the $L1\ Loss$, with the latent code $w$ serving as the label.

$$\mathcal{L}_{reg} = \mathbb{E}_{w \sim p(z)} \|\hat{w} - w\|_1 \tag{6}$$

As illustrated in Figure 5, the Latent Regularizer is capable of eliminating the random noise introduced into the latent codes, thereby correcting the outlier latent codes back into the reasonable distribution of the $\mathcal{W}^+$ space. Indeed, through the reconstruction task, the Latent Regularizer learns to (i) infer latent codes from their internal structure, and (ii) restore erroneous and missing data. This process facilitates the Latent Regularizer in learning the mapping representations of natural image prior distributions in the $\mathcal{W}^+$ space.

## 3.3 Latent Predictor

This section will elaborate on the proposed Latent Predictor, as well as the joint training of the Latent Predictor and the Latent Regularizer.

As illustrated in Figure 4 (b), the mapping network of StyleGAN2 [20] randomly samples a 512-dimensional latent code $z$ from a normal distribution and maps it to a latent code $w_0$ of dimension $l \times 512$, where $l$ represents the number of layers in the generator network. Subsequently, we slightly perturb the first six layers of $w_0$ to obtain $w_1$, and then similarly perturb the first six layers of $w_1$ to acquire $w_2$, and so forth. By repeating this process of minor random perturbations n times, we generate a sequence of latent codes $w_0, w_1, w_2, \ldots, w_n$. The perturbation process is as follows:

$$w_i = w_{i-1} - \lambda \cdot (w^* - w_{i-1}) \tag{7}$$

where $\lambda$ is a constant, $w^*$ is an independently sampled latent code unrelated to $w_0$, and $i = 1, 2, \ldots, n$.

Prior work [8, 17–20, 29] finds that manipulating the first six layers of the latent codes enables spatial control, such as pose, orientation and shape. Our perturbation process does not affect other styles, such as color and texture. Therefore, our perturbation facilitates the preservation of identity information during the image dragging process. As illustrated in Figure 6, our perturbation process of latent codes in the StyleGAN2 [20] latent space corresponds to spatial variations in the pixel space of images, such as pose, orientation, and shape. Therefore, the sequence $w_0, w_1, \ldots, w_n$ is a latent code motion sequence. By utilizing this motion sequence as a training sample, the image dragging problem can be decomposed into multiple sequential sub-problems. Between two consecutive sub-problems, the majority of pixels in the images before and after dragging remain consistent, requiring the model to predict the movement of only a small portion of pixels, thereby significantly reducing the complexity of the problem.

The Latent Predictor employs a straightforward teacher-forcing cross-attention Transformer Decoder [46] for motion sequence prediction. The latent codes $w_0$ and $w_n$ are processed through the StyleGAN2 generator network [20] to produce the synthesized images $I_0$ and $I_n$, respectively. An off-the-shelf feature matching algorithm [2] is applied to $I_0$ and $I_n$, with matching points whose pixel distance exceeds 50 selected as training sample points. The matching points of $I_0$ are designated as handle points (for instance, the red point of $I_0$ in Figure 4 (b)), and those of $I_n$ as target points (for instance, the blue point of $I_n$ in Figure 4 (b)). DragGAN [29] focuses on the feature maps from the 6th block of StyleGAN2 [20], as they offer an optimal balance between resolution and discriminative power, outperforming other features in effectiveness. Inspired by DragGAN [29], we use the feature map obtained after passing $w_0$ through the 6th block of the StyleGAN2 [20] generator network as the intermediate feature map $F_0$ in our work. Subsequently, we extract small patches corresponding to the positions of handle points and target points on $F_0$. After $F_0$ undergoes convolution to extract spatial information, it is concatenated with the small patches to serve as the key and value for the cross-attention mechanism. The sequence composed of $w_0, w_1, \ldots, w_{n-1}$ is combined with position embeddings through element-wise addition, serving as the query for the cross-attention mechanism. During training, teacher forcing is employed to predict $\hat{w}_1, \hat{w}_2, \ldots, \hat{w}_n$. The final output is connected via skip connection to the Latent Regularizer, to constrain the predicted latent code motion sequences within the reasonable distribution of the $\mathcal{W}^+$ space.

$$k = K \cdot MLP([MLP(F_{seq}), MLP(P_{seq})]) \tag{8}$$

$$v = V \cdot MLP([MLP(F_{seq}), MLP(P_{seq})]) \tag{9}$$

$$q = Q \cdot (MLP([w_0; w_1; \ldots; w_{n-1}]) \oplus PE) \tag{10}$$

$$[\hat{w}_1; \hat{w}_2; \ldots; \hat{w}_n] = D(MLP(softmax(q \cdot k^T) \cdot v)) \tag{11}$$

where $F_{seq}$ denotes the sequence of feature vectors extracted from the intermediate feature $F_0$ after spatial convolution to gather information, and $P_{seq}$ represents the 7x7 patches at the positions

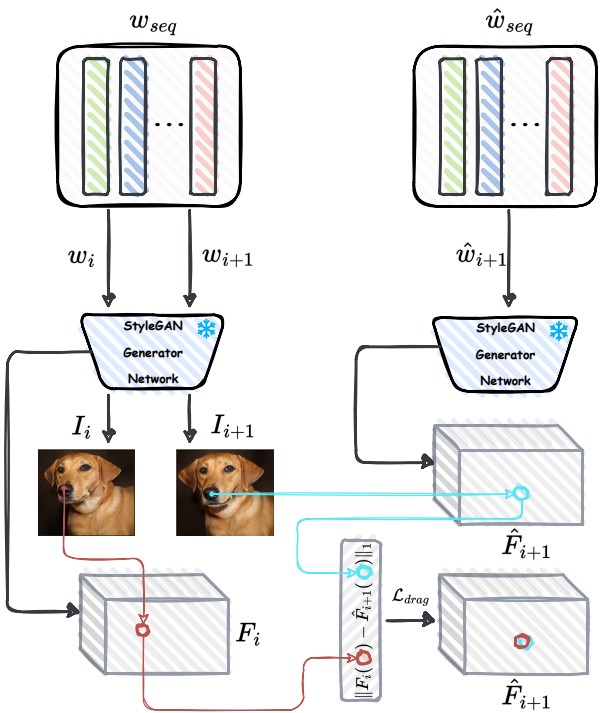

Figure 7: Drag Loss. The Drag loss supervises the patches of intermediate features to guide the handle points towards the target points.

corresponding to the handle points and target points on $F_0$, and $PE$ denotes the position embedding. The notation $[;]$ is used to represent the formation of a latent code sequence, while $[,]$ indicates the concatenation operation.

The Latent Predictor aims to learn the state transition path from $w_0$ to $w_n$, with the L1 loss function employed as the loss function.

$$\mathcal{L}_{pred} = \mathbb{E}_{w \sim p(z)} \|\hat{w}_{seq} - w_{seq}\|_1 \qquad (12)$$

where $\hat{w}_{seq}$ denotes the set consisting of $w_0, \hat{w}_1, \hat{w}_2, \dots, \hat{w}_n$, and $w_{seq}$ represents the set containing $w_0, w_1, \dots, w_{n-1}, w_n$.

Furthermore, we apply a drag loss to the intermediate feature maps to guide the handle points towards the target points. Specifically, we use the feature of the handle patch before dragging as supervision for the feature of the target patch after dragging.

$$\mathcal{L}_{drag} = \sum_{i=0}^{n-1} \sum_{j=1}^{m_i} \sum_{\substack{h_{i,j} \in \Omega(H_{i,j}) \\ t_{i,j} \in \Omega(T_{i,j})}} \|F_i(h_{i,j}) - \hat{F}_{i+1}(t_{i,j})\|_1 \qquad (13)$$

where $n$ represents the length of the latent code motion sequence, and $m_i$ is the number of matching points between the generated images $I_i$ and $I_{i+1}$ corresponding to $w_i$ and $w_{i+1}$, with only matching points exceeding a pixel distance of 30 being selected. The matching points on $I_i$ are designated as handle points $H_{i,j}$ (the red point of $I_i$ in Figure 7), and those on $I_{i+1}$ are designated as target points $T_{i,j}$ (the blue point of $I_{i+1}$ in Figure 7). We use $\Omega(H_{i,j})$ to represent

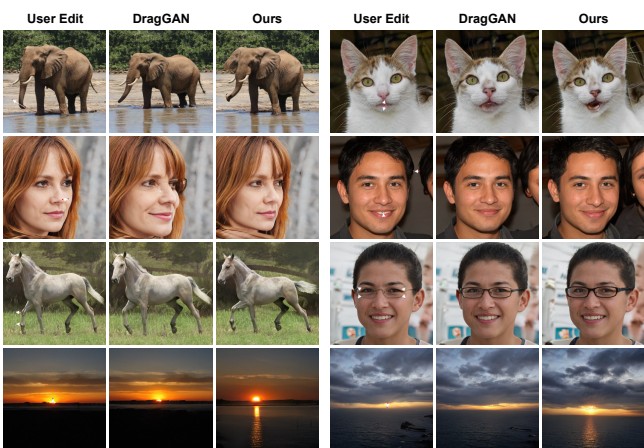

Figure 8: A qualitative comparison of the image editing performance between our method and DragGAN [29].

the pixels within a 7x7 patch centered at $H_{i,j}$. $F_i$ and $\hat{F}_{i+1}$ are the intermediate feature maps of $w_i$ and $\hat{w}_{i+1}$, respectively. $F(h_{i,j})$ denotes the feature values of $F$ at pixel $h_{i,j}$. This loss function encourages the handle points to move towards the target points.

Finally, the overall loss function is defined as:

$$\mathcal{L} = \alpha \mathcal{L}_{pred} + \beta \mathcal{L}_{drag} \qquad (14)$$

where $\alpha$ and $\beta$ are coefficients to balance the two loss functions, with $\alpha$ set to 0.1 and $\beta$ set to 1 by default in our experiments.

## 4 EXPERIMENT

### 4.1 Training And Inference

Following the setup of DragGAN [29], we utilized the StyleGAN2 [20] pre-trained on the following datasets (the resolution of the pretrained StyleGAN2 [20] is shown in brackets): FFHQ (512) [19], AFHQCat (512) [4], SHHQ (512) [10], LSUN Car (512) [48], LSUN Cat (256) [48], Landscapes HQ (256) [40] and self-distilled dataset from Self-distilled stylegan [28] including Lion (512) [28], Dog (1024) [28], and Elephant (512) [28].

The Latent Regularizer employs a standard transformer architecture, consisting of a self-attention mechanism with 6 transformer encoder layers, and a cross-attention mechanism with 6 transformer decoder layers [46]. The Latent Predictor consists of a self-attention mechanism with 6 transformer encoder layers, and a cross-attention mechanism with 16 transformer decoder layers [46].

In the first stage of training, only Latent Regularizer requires training, with its learning rate set to $1 \times 10^{-3}$. In the second stage of training, both Latent Regularizer and Latent Predictor require joint training. The learning rate for Latent Regularizer is set at $1 \times 10^{-5}$, while Latent Predictor employs a cosine annealing decay for its learning rate, with an initial value set at $1 \times 10^{-5}$, a minimum value at $1 \times 10^{-7}$, and a decay period of 30. The first stage of training requires 50 epochs. The second stage of training requires 150 epochs. The mapping network and generator network of StyleGAN2 [20] are both frozen.

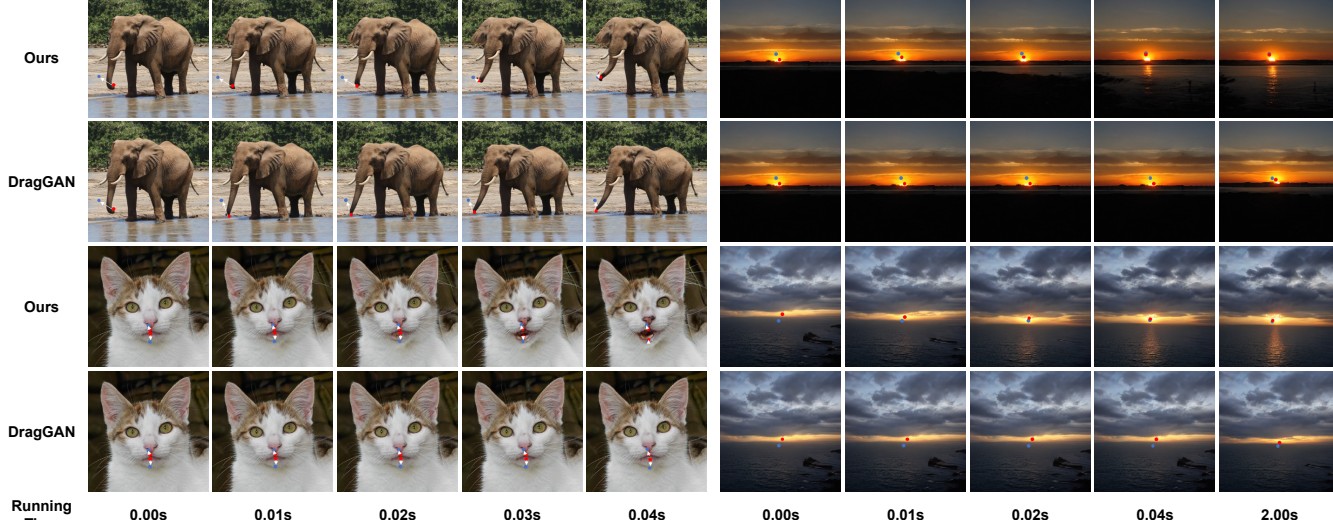

**Figure 9: A qualitative comparison between our method and DragGAN [29] in terms of inference speed and image editing performance.**

The user inputs handle points and target points on the initial image, which are then processed through the Latent Predictor and the Latent Regularizer, resulting in the edited image.

## 4.2 Qualitative Evaluation

Figure 1 illustrates the comparison between our method and Drag-GAN [29] under complex editing scenarios, while Figure 8 displays the comparison in simple editing scenarios. Figure 9 shows a comprehensive comparison of editing speed and image editing performance between our method and DragGAN [29]. Our method all outperforms DragGAN [29].

## 4.3 Quantitative Evaluation

Following the setup of DragGAN [29], we conducted a quantitative evaluation of our method, encompassing facial landmark manipulation and paired image reconstruction.

*4.3.1 Face Landmark Manipulation.* Following the setup of Drag-GAN [29], we employed an off-the-shelf tool, Dlib-ml [22], for facial landmark detection. Subsequently, we utilized a StyleGAN2 [20] pre-trained on the FFHQ [19] dataset to randomly generate two facial images, upon which we performed landmark detection. Our objective is to manipulate the landmarks of the first facial image to align them with the landmarks of the second facial image. Subsequently, we calculate the mean distance (MD) between the landmarks of the two images. The results are derived from an average of 2000 tests using the same set of test samples to evaluate all methods. In this manner, the final Mean Distance (MD) score reflects the efficacy of the method in moving the landmarks to the target positions. Evaluations were conducted with varying numbers of landmarks, including 1, 5, and 68. Additionally, we report the Frechet Inception Distance (FID) scores between the edited images and the initial images.

**Table 1: Quantitative evaluation on face landmark manipulation. We calculate the mean distance (MD) between the landmarks of the two images. The FID and Time are reported based on the '1 point' setting. Red font indicates the best performance, while blue signifies the second best.**

| Method | 1 point | 5 points | 68 points | FID | Time (s) |
|---|---|---|---|---|---|
| No edit | 14.76 | 12.39 | 15.27 | - | - |
| UserControllableLT | 11.64 | 10.41 | 10.15 | 25.32 | **0.03** |
| FreeGAN | **1.45** | **3.03** | **4.17** | **7.67** | 2.0 |
| DragGAN | 1.62 | 3.23 | 4.32 | 8.30 | 1.9 |
| Ours w/o Latent Regularizer | 3.75 | 5.79 | 11.14 | 17.23 | 0.12 |
| Ours w/o Latent Predictor | 4.94 | 12.78 | 25.63 | 26.34 | 0.15 |
| Ours | **1.33** | **3.02** | **3.56** | **6.46** | **0.04** |

The results are provided in Table 1. Our method achieves performance comparable to DragGAN [29] under different numbers of points. According to the FID scores, the image quality post-editing with our approach is superior. In terms of speed, our method significantly surpasses DragGAN [29]. Overall, our results are comparable to DragGAN [29], but with a faster execution speed.

*4.3.2 Paired Image Reconstruction.* In our study, both our method and DragGAN [29] were evaluated using the same settings as those employed in UserControllableLT [8]. In this study, we begin with a latent code $w_1$ and apply random perturbations to it in the same manner as described in UserControllableLT [8], thereby generating another latent code $w_2$. Subsequently, we use these two latent codes to generate StyleGAN2 [20] images $I_1$ and $I_2$, respectively. Following this, we calculate the optical flow between $I_1$ and $I_2$ and randomly select 32 pixel points from the flow field as user input $U$. Our research objective is to reconstruct image $I_2$ using only $I_1$ and $U$. The results are provided in Table 2. In most datasets, our approach demonstrates superior performance compared to DragGAN [29].

*4.3.3 Ablation Study and Analysis Experiment.* In this context, we investigate the roles of the Latent Regularizer and Latent Predictor

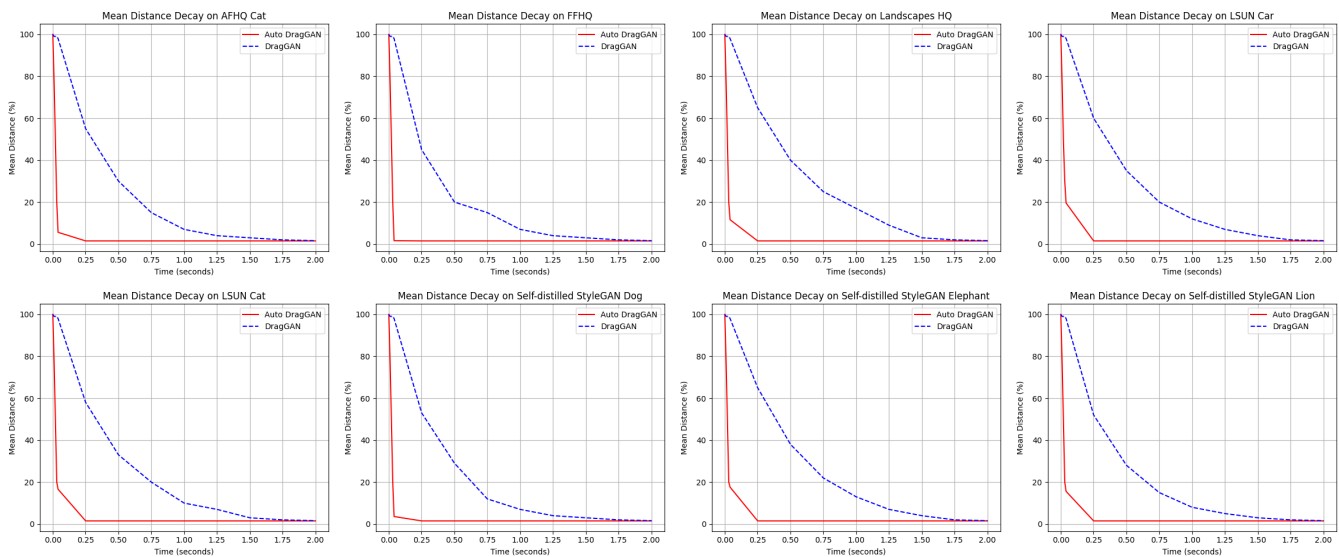

**Figure 10: A comparative analysis of the Mean Distance Decay between Auto DragGAN and DragGAN [29] across various datasets. Auto DragGAN demonstrates faster convergence compared to DragGAN [29]. The convergence point at 2s indicates that both methods have comparable abilities in dragging the handle points to the target points.**

in influencing the performance of the model. The results are provided in Table 1 and Table 2. 'Ours w/o Latent Regularizer' denotes the scenario where the Latent Regularizer is not utilized for the restoration of the latent codes. 'Ours w/o Latent Predictor' denotes the scenario where the length of latent code motion sequence, $n$, equals 1. Additionally, we discuss the impact of $n$ on the effectiveness of the model trained in the final stage. The results are provided in Table 3.

*4.3.4 Mean Distance Decay.* We introduce a new metric, **Mean Distance Decay (MDD)**, to assess the speed performance of drag editing. MDD represents the ratio of the current mean distance between the handle points and the target points to the initial mean distance. A smaller value indicates a closer proximity between the handle points and the target points. The formula is as follows:

$$MDD = \frac{MD_{cur}}{MD_{init}} \tag{15}$$

where $MD_{cur}$ represents the mean distance at the current moment, while $MD_{init}$ denotes the initial mean distance.

As illustrated in the Figure 10, our approach demonstrates faster convergence in drag operations across various datasets compared to DragGAN [29]. Our method begins to converge at approximately **0.04s** across various datasets, whereas DragGAN [29] starts to converge around **2s**. The calculation of **Mean Distance Decay (MDD)** is based on selecting only a single pair of handle point and target point. We calculated the **Mean Distance Decay (MDD)** on each dataset. We selected a handle point and a target point on the initial image, and then performed a drag operation to calculate the **Mean Distance Decay (MDD)** for both methods. As illustrated in the Figure 10, the final experimental results indicate that our method converges more rapidly than DragGAN [29], while maintaining a comparable dragging capability with DragGAN [29].

**Table 2: Quantitative evaluation on paired image reconstruction. We follow the evaluation in UserControllableLT [8] and report MSE ($\times 10^2$)↓ and LPIPS ($\times 10$)↓ scores. Red font indicates the best performance, while blue signifies the second best.**

| Dataset | Lion | | LSUN Cat | | Dog | | LSUN Car | |
|---|---|---|---|---|---|---|---|---|
| Metric | MSE | LPIPS | MSE | LPIPS | MSE | LPIPS | MSE | LPIPS |
| UserControllableLT | 1.82 | 1.14 | 1.25 | 0.87 | 1.23 | 0.92 | 1.98 | 0.85 |
| DragGAN | 0.52 | 0.70 | 0.88 | 0.86 | 0.39 | 0.42 | 1.75 | 0.77 |
| FreeGAN | 0.48 | 0.67 | 0.79 | 0.96 | 0.38 | 0.37 | 1.64 | 0.64 |
| Ours w/o Latent Regularizer | 1.52 | 1.38 | 1.33 | 0.83 | 0.94 | 0.83 | 2.01 | 0.94 |
| Ours w/o Latent Predictor | 1.74 | 1.12 | 1.51 | 0.95 | 1.97 | 0.93 | 2.13 | 0.98 |
| Ours | 0.42 | 0.58 | 0.70 | 0.63 | 0.31 | 0.32 | 1.53 | 0.58 |

**Table 3: Effects of $n$. Paired image reconstruction on Dog dataset. We follow the evaluation in UserControllableLT [8] and report MSE ($\times 10^2$)↓ score.**

| $n$ | 1 | 2 | 3 | 4 | 5 | 7 | 9 | 10 |
|---|---|---|---|---|---|---|---|---|
| MSE | 1.97 | 1.18 | 0.68 | 0.54 | **0.37** | 0.39 | 0.39 | 0.38 |

## 5 CONCLUSION

We propose Auto DragGAN, which, unlike DragGAN [29], Free-Drag [26], and DragDiffusion [39] that optimize latent vectors, is an autoregression-based model we have developed to learn the movement paths of latent codes within the latent space. Our method benefits from learning the variation patterns of latent codes during the image dragging process, and it significantly outperforms other methods [26, 29, 39] in handling complex dragging scenarios. This approach not only matches but slightly exceeds the effectiveness of DragGAN [29] while significantly boosting processing speed.

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
