# OpenReview forum: "Auto DragGAN: Editing the Generative Image Manifold in an Autoregressive Manner"
_acmmm.org/ACMMM/2024/Conference — MM2024 Poster_

### Official Review · Reviewer_Umji · 2024-05-16

**Rating:** 5
**Confidence:** 2

**Summary:**

This paper introduces a regression-based network that efficiently learns StyleGAN latent code variations for precise pixel-level edits. Users specify handle and target points, and the method moves the handle points to target points with minimal time cost. An encoder-decoder 'Latent Predictor' manages these movements, while a 'Latent Regularizer' ensures they remain within natural image distributions. The approach achieves state-of-the-art speed and precision in image editing.

**Strengths:**

1. I really like the contribution of this paper, including a Latent Predictor and a Latent Regularizer. From my point of view, the idea of delivering regression-based to accelerate Drag-GAN is promising and novel.
2. The result of this paper is pretty cool, including accelerating Drag-GAN a lot and also improving the image fidelity.
3. The paper is basically well-written and well-constructed.

**Limitations:**

1. The paper fails to discuss the effects of some hyper-parameters. For example, when introducing random noise to 𝑤, the authors set the vector values to zero with a 25% probability, followed by the addition of Gaussian noise. Why do you choose such probability? If the probability is too high or too low, the Latent Regularizer may not be able to restore the erroneous and missing data.
2. Could you please provide some explanation of why w1' can be restored from w2'? What if w2' also contains some noise or missing data?
3. Have you ever compared the inference memory cost of your method and Drag-GAN? Compared with Drag-GAN, this method needs a large Transformer as the Latent Regularizer and another large Transformer as the Latent Predictor. Therefore, I guess it requires more memory cost than Drag-GAN in the inference phase.
4. The total training time and hardware requirements are missing.

**Suitability:**

2

---

### Official Review · Reviewer_rmky · 2024-05-24

**Rating:** 3
**Confidence:** 3

**Summary:**

The paper introduces Auto DragGAN for ideal trade-off between control granularity and inference speed. With designed Latent Regularizer and Latent Predictor, Auto DragGAN achieves state-of-the-art performance in inference speed and image editing.

**Strengths:**

1. The idea is novel and interesting. In this paper, the authors propose Auto DragGAN which is the first work to convert the image dragging problem into a regression problem. And with the novel designs, Auto DragGAN achieves the SOTA performance.
2. The paper is well-written and easy to follow.

**Limitations:**

1. The motivation is unclear. The authors should explain why regression-based network can achieve pixel-level fine-grained image editing more effectively than the previous methods.

2. Concerns about the visual comparisons of image editing in Fig.1. Although Auto DragGAN precisely moves the handle points to the target points, the shape of an object is changed meanwhile. For example, the hair of 'lion' in FIg.1 on its head undergoes significant changes.

**Suitability:**

3

---

### Official Review · Reviewer_Caaa · 2024-05-28

**Rating:** 4
**Confidence:** 2

**Summary:**

The paper proposes an approach to pixel-level, fine-grained image editing using generative adversarial networks (GANs). By employing a regression-based network, the method allows for precise image editing by dragging specified points on the image to desired target points. The core innovation lies in the 'Latent Predictor' and 'Latent Regularizer', which work together to predict and correct the movement of latent codes in the GAN's manifold, ensuring high fidelity and speed in image editing.

**Strengths:**

The method seems to achieve a superior trade-off between control granularity and inference speed. The paper provides extensive experimental results demonstrating the method's state-of-the-art performance in terms of both speed and image fidelity.

**Limitations:**

- It seems to me that the performance of the proposed approach heavily depends on the precise selection of handle points and target points.

- I have concerns regarding the efficiency of the proposed approach, given the dual-stage training process and the need for specific components. The author are suggested to include an efficiency analysis.

- The authors are encouraged to include more comparison results with diffusion-based approaches, in terms of quality and efficiency.

**Suitability:**

3

---

### Meta-Review · Area_Chair_k663 · 2024-07-04

**Recommendation:** Accept (Poster)
**Confidence:** 4

**Metareview:**

The manuscript has received three reviews, based on which a rebuttal was submitted.

After the rebuttal, all the reviewers are on the positive side, indicating the manuscript meets the bar of MM.

As such, there is no basis to overturn the consensus. Congrats!